# Workup and Clinical Assessment for Allergen Immunotherapy Candidates

**DOI:** 10.3390/cells11040653

**Published:** 2022-02-14

**Authors:** Constantinos Pitsios, Konstantinos Petalas, Anastasia Dimitriou, Konstantinos Parperis, Kyriaki Gerasimidou, Caterina Chliva

**Affiliations:** 1Medical School, University of Cyprus, Nicosia 2109, Cyprus; parperis.konstantinos-marinos@ucy.ac.cy; 2Department of Allergy and Clinical Immunology, 251 General Airforce Hospital, 155 61 Athens, Greece; kpetalas@hotmail.com; 3Allergy Private Practice Network, 115 27 Athens, Greece; drdamiana@gmail.com; 4Medical School, National and Kapodistrian University of Athens, 115 27 Athens, Greece; gerkorina@gmail.com; 5Allergy Unit, 2nd Department of Dermatology and Venereology, “Attikon” General University Hospital, 124 61 Haidari, Greece; cchliva@gmail.com

**Keywords:** allergen immunotherapy, venom immunotherapy, contraindications, allergy diagnosis

## Abstract

Allergen Immunotherapy (AIT) is a well-established, efficient, and safe way to treat respiratory and insect-venom allergies. After determining the diagnosis of the clinically relevant culprit allergen, AIT can be prescribed. However, not all patients are eligible for AIT, since some diseases/conditions represent contraindications to AIT use, as described in several guidelines. Allergists are often preoccupied on whether an extensive workup should be ordered in apparently healthy AIT candidates in order to detect contra-indicated diseases and conditions. These preoccupations often arise from clinical, ethical and legal issues. The aim of this article is to suggest an approach to the workup and assessment of the presence of any underlying diseases/conditions in patients with no case history before the start of AIT. Notably, there is a lack of published studies on the appropriate evaluation of AIT candidates, with no globally accepted guidelines. It appears that Allergists are mostly deciding based on their AIT training, as well as their clinical experience. Guidance is based mainly on experts’ opinions; the suggested preliminary workup can be divided into mandatory and optional testing. The evaluation for possible underlying neoplastic, autoimmune, and cardiovascular diseases, primary and acquired immunodeficiencies and pregnancy, might be helpful but only in subjects for whom the history and clinical examination raise suspicion of these conditions. A workup without any reasonable correlation with potential contraindications is useless. In conclusion, the evaluation of each individual candidate for possible medical conditions should be determined on a case-by-case basis.

## 1. Introduction

Allergen Immunotherapy (AIT) is a well-established treatment option for respiratory and insect-venom allergies, as well as the only etiology-based and disease-modifying treatment for allergic diseases [1,2,3,4]. The administration of AIT is indicated for the treatment of allergic rhinitis, allergic rhinoconjunctivitis and allergic asthma, and is efficacious when directed at the specific allergen-driving symptoms [1,3,4,5,6]. Venom immunotherapy (VIT) is recommended for children and adults following a systemic allergic reaction to insect stings that exceeds generalized cutaneous symptoms [6,7,8].

Allergic reactions induced by AIT administration are common adverse events. The frequency of systemic adverse events is 8–20% for patients receiving VIT and 2.1% for patients (or 0.2% of the injections) receiving subcutaneous immunotherapy (SCIT) for airborne allergens [6,7,8,9]. The sublingual route of AIT for respiratory allergies (SLIT) appears to be a safer option with only 1.1% of patients reporting adverse events [8,9]. Local reactions at the injection site, that resolve spontaneously, are the main side effect of subcutaneous AIT, ranging from 0.7–4% of the injections [3,6]. Local reactions at the oral mucosa can be observed in the case of SLIT [3]. Although systemic allergic reactions are a significant concern during AIT, the risk of anaphylaxis can be minimized when well-trained allergologists and other healthcare professionals follow standard protocols [6]. Further, the prompt recognition and management of the potential reaction is a crucial step [3,4,6].

Although acute reactions remain the most common side effects of AIT, close attention should be paid to slowly evolving reactions that might appear as an interaction of AIT with the underlying pathophysiological mechanism of certain health conditions. The administration of AIT is contraindicated in cases where certain concomitant diseases or conditions are present; however, controversy exists on what the contraindications are, on the level of evidence for their consideration as contraindications and on whether they are relative or absolute [10,11]. On the other hand, the administration of AIT appears to be safe and well tolerated in certain patient groups with high-risk conditions such as cardiovascular disease, those undergoing treatment with ACE inhibitors or beta blockers, and patients with malignant diseases in remission or organ-specific autoimmune diseases, [11].

National and international guidelines are valuable tools for clinicians, describing the indications and contraindications of AIT, although differences might exist among them [10]. Following the identification of the culprit antigen that correlates with the suspected triggers and patient exposure, AIT can be prescribed to the patient [5,7]. A thorough history and clinical examination can provide information regarding contraindications.

It is worth noting that several clinicians and institutions follow the practice of ordering screening tests based on conditions that are considered contraindications. The practice of ordering a battery of tests is not supported by current guidelines and is based upon the clinician’s discretion. Confirmation of the positive allergy skin tests by serum-specific IgE (sIgE) is an example of a workup that is practiced by many allergy centers before prescribing initial AIT, although the guidelines do not suggest it [5,6]. A complete blood count and peripheral smear examination are common laboratory tests used as a routine health checkup of a healthy person and are also used by several institutions as a screening tool for AIT candidates.

In this paper, an overview of the preliminary AIT workup to assess the presence of certain underlying conditions that are considered as contraindications is described. Given the fact that the evidence supporting this approach is limited, clinical judgement and shared decision making is warranted.

## 2. Tests Confirming IgE-Mediated Allergy and Monitoring the Severity of the Allergic Reaction

### 2.1. Skin Tests

Skin tests are the cornerstone of allergy diagnostic evaluation. The skin-prick test (SPT) is a cheap, quick and easy-to-perform method of diagnosis for IgE-mediated sensitivity to aeroallergens and Hymenoptera venoms [12]. In the case of Hymenoptera-venom allergy, the SPT is followed by the performance of intradermal tests [13]. Regarding the respiratory allergy evaluation, the SPT is performed using a panel of standard allergen extracts, including the local major aeroallergens [12]. In the case of venom hypersensitivity, skin tests are performed with the use of locally offending insects. In Europe, *Apis melifera, Vespula, Polistes* and *Dolichovespula* venom extracts are widely used [13].

### 2.2. Serum-IgE Tests

The published guidelines recommend the use of serum sIgE as a useful test under certain circumstances or as an alternative to the SPT [5,6,7]. There is no doubt that the use of both the SPT and sIgE increases the diagnostic sensitivity [14]. Two-tiered allergen testing by two independent diagnostic tools can increase the confidence in the long-term success of the AIT. The sIgE tests have a “quantitative” value and can replace the SPT in cases of extended dermatosis, in patients taking histamine-blocking drugs, or in non-cooperative children [15].

Cautiousness should be paid to the interpretation of sIgE in the case of high total serum IgE; in this case, the detection of low specific-IgE levels is often of doubtful clinical relevance [12]. Combined with total IgE, the use of sIgE has also been proposed as a predictive biomarker for the efficacy of AIT, but its utility has not been properly evaluated or validated [16].

### 2.3. Component-Resolved Diagnostics (CRD)

Allergen cross-reactivity, defined as the immunologic recognition of different antigens by the same IgE, is a frequent phenomenon observed among the pollen of taxonomically related plants [15]. Cross-reactivity is also observed for homologous molecules that are widely distributed in evolutionarily unrelated species, namely panallergens. When patients are skin- and sIgE-tested, cross-reactivity with the false-positive tests of homologous allergens without clinical relevance may occur [17]. A polysensitized patient is not always poly-allergic; polysensitization is the presentation of multiple positive (sensitivities to) allergy tests, while a poly-allergic patient is also polysensitized but with clinically relevant positive sensitivities. This phenomenon can be observed in both airborne and Hymenoptera-venom allergens, so in the case of polysensitized patients, caution should be paid to detect whether it is a true co-sensitization (true coexisting sensitization to different allergens) and to exclude sensitization to a cross-reactive allergen that is not connected to the clinical symptoms.

Recent advances in molecular allergology have provided the opportunity to use component-resolved diagnostics (CRD) to meticulously interpret the allergy tests. CRD offers the possibility to detect “truly symptom-causing” allergen molecules, called marker allergens, that are specific to a pollen or a Hymenoptera-species venom [13]. CRD is often necessary to exclude false-positive tests.

An example of CRD as an additional decision tool is its use in grass AIT; the detection of sIgE with any of *Phl p 1, Phl p 2, Phl p 5* or *Phl p 11* leads to the safe initiation of AIT, while the co-sensitization to *Phl p 5* and *Phl p 12* predicts side effects during AIT [17,18]. Examples of allergens associated with cross-sensitivity to Apis mellifera and Vespula vulgaris are hyaluronidases, dipeptidylpeptidase IV and vitellogenins [13]. Sensitization to these allergens can lead to false-positive allergy tests. CRD can also reveal sIgE against cross-reactive carbohydrate determinants that occur in patients sensitized to pollen or venoms without clinical relevance [13].

### 2.4. Tryptase

Patients with mastocytosis, particularly those with clonal-mast-cell-activation syndrome (c-MCAS), are at high risk for anaphylaxis after a field sting [19]. However, patients with aggressive subtypes of systemic mastocytosis and those with urticaria pigmentosa appear not to be at risk of a systemic sting reaction [19,20]. Tryptase is a useful diagnostic tool that is included as a minor criterion in the diagnostic criteria for systemic mastocytosis [21].

A history of severe Hymenoptera-venom anaphylaxis in patients with c-MCAS is predictive of a future severe systemic sting reaction, and VIT is the appropriate therapeutic option regardless of the level of tryptase. A VIT duration longer than the usual 5 years, or even a lifelong duration, is highly advised for these patients [7]. Increased serum-tryptase levels are associated with more frequent and severe systemic reactions to VIT injections, greater treatment-failure rates during VIT treatment and greater relapse rates, including fatal reactions, if VIT is discontinued [6].

The measurement of baseline tryptase is recommended in patients with moderate or severe anaphylactic reactions to stings, in order to detect mastocytosis. It may represent a predictive factor of VIT efficacy and affect the decision regarding the treatment duration [7]. However, elevated tryptase levels may represent an epiphenomenon of the enhanced mast-cell activation and/or relatively increased mast-cell numbers [19].

### 2.5. Basophil-Activation Test

The basophil-activation test (BAT) is a useful technique to establish a diagnosis in several allergy cases for which the common diagnostic tools have failed to accurately identify the culprit allergen. For example, the BAT can be useful in determining a diagnosis in patients with a history of systemic sting reactions, with negative skin and sIgE tests and with a hint of possible mastocytosis [22]. Therefore, it can only be a preliminary-workup diagnostic tool for VIT in rare cases of sting-induced anaphylaxis [23].

### 2.6. Complete Blood Count (CBC)

The CBC includes a hemogram with the enumeration of red blood cells (RBCs), white blood cells (WBCs) and platelets. It is a useful test to evaluate primary diseases of the blood and bone marrow, including anemia, leukemia, polycythemia, thrombocytosis and thrombocytopenia [24]. Furthermore, it is used in the evaluation of disease processes such as infection, inflammation, coagulopathies, neoplasms and exposure to toxic substances [24].

In allergic patients, a CBC with a WBC differential often reveals mild eosinophilia (500–1500 eosinophils per mL) [25]. This is a common finding in patients with atopic dermatitis, asthma and drug-hypersensitivity reactions and no further detection is needed in order to start AIT. Hypereosinophilia (>1500 eos per mL) should be differentiated from eosinophilia, given that it is usually observed in parasitic infections and hypereosinophilic syndromes and rarely in drug allergies.

Primary and acquired immunodeficiencies (IDs) are considered relative contraindications for AIT. Several primary IDs are associated with eosinophilia and hypereosinophilia, such as autosomal-dominant hyper-IgE (or Job’s) syndrome, Omenn syndrome, Wiskott–Aldrich syndrome and Severe Combined Immunodeficiency due to adenosine deaminase deficiency (ADA-SCID) [26]. Given the characteristic clinical features and the history of recurrent infections, most patients with primary IDs are diagnosed early in life, therefore, ID is low on the differential-diagnosis list of eosinophilia. Eosinophilia might also occur in immune-dysregulatory syndromes, autoimmune lymphoproliferative syndrome, X-linked syndrome with immunodysregulation, polyendocrinopathy and enteropathy, Loeys–Dietz syndrome and in dermatologic syndromes with immunodysregulation [26].

In conclusion, the evaluation of CBC is an important priority before initiating AIT, since it can detect malignancies as well as medical conditions requiring treatment. On the other hand, the detection of mild eosinophilia in the WBCs of patients with respiratory allergy does not require further blood tests.

## 3. Cardiovascular Checkup

Cardiovascular diseases do not consist of a contraindication to AIT [7,27]. On the contrary, VIT is recommended for venom-allergic patients with a history of coronary heart disease, since a future episode of sting-induced anaphylaxis can impair coronary blood flow, significantly contributing to an unfavorable outcome [28,29]. A large European survey on systemic reactions during respiratory AIT revealed that previous cardiovascular disease does not constitute a risk factor for anaphylaxis [9]

Anaphylactic reactions are frequently associated with transient alteration in cardiovascular function, but in some cases they may result in extensive and life-threatening myocardial damage. Cardiovascular manifestations of anaphylaxis include hypotension and shock, cardiac arrhythmias, ventricular dysfunction, and cardiac arrest [30]. These symptoms may be partly due to vasodilatation and increased vascular permeability leading to hypovolemia and partly to the direct cardiotoxic effect of mast cell mediators and hypoxemia following bronchospasm and shock. In addition, acute ischemic events, including angina and myocardial infarction, are currently considered as part of the clinical picture of anaphylaxis [31]. The World Allergy Organization’s (WAO) guidelines for the management of anaphylaxis recognize cardiovascular diseases as an important patient-related factor that are associated with an increased risk of severe or fatal anaphylactic episodes [32]. These data underline the protective value of VIT in patients with cardiovascular diseases.

Epinephrine is the treatment of choice for anaphylaxis, even if a history of cardiovascular disease exists. Epinephrine increases vasoconstriction, peripheral vascular resistance, and blood pressure through alpha-1 adrenergic-receptor stimulation, thereby preventing and relieving life-threatening hypotension, shock, laryngeal edema, and upper-airway obstruction [32]. Through beta-1 adrenergic-receptor stimulation, epinephrine exhibits inotropic and chronotropic effects. It also promotes bronchodilation through beta-2 adrenergic receptors [33], and when used promptly, it suppresses the release of mediators from mast cells and basophils [34].

For patients taking b-blockers, there is a theoretical risk that anaphylaxis can be more severe and refractory to treatment with epinephrine. Angiotensin-converting-enzyme (ACE) inhibitors directly interfere with the metabolism of bradykinin, as the ACE is a key enzyme responsible for its degradation. Moreover, ACE inhibitors and angiotensin-receptor blockers (ARBs) could potentially impair the endogenous compensatory response of the renin-angiotensin system, which is crucial to maintaining peripheral vasoconstriction in the case of severe hypotension [35].

According to a multicenter Emergency-Department study, the use of antihypertensive drugs may influence the outcome of anaphylaxis; the use of beta-blockers, ACE inhibitors, ARBs, calcium-channel blockers and diuretics is associated with increased organ-system involvement and increased odds of hospital admission, independently of age, sex, suspected trigger or pre-existing lung disease [36]. On the other hand, evidence to support that beta-blockers increase the rate or the risk of severe reactions during immunotherapy is relatively weak [37,38].

SCIT is associated with a lower risk of developing ischemic heart disease and acute myocardial infarction compared to conventional allergy treatment [39]. The risk of near-fatal and fatal anaphylaxis during the administration of SCIT remains quite low but requires physicians to be aware of it. On the other hand, anaphylactic events are uncommon in SLIT.

There is an ongoing debate on whether emergency treatment used to treat a reaction during SCIT could be effective in patients treated with beta-blockers, and also on whether treatment with beta-blockers or ACE inhibitors is a risk factor for more severe or more frequent side effects during VIT. The AAAAI guidelines suggest that “concomitant use of beta-blockers and AIT should be carefully considered from an individualized risk/benefit standpoint” and “concurrent administration of VIT and an ACE inhibitor is warranted in select cases in which no equally efficacious alternative for an ACE inhibitor exists” [6]. In the EAACI guidelines, the use of beta-blockers is suggested as a relative contraindication for SCIT and SLIT, while the use of ACE inhibitors is a relative contraindication for VIT [27].

Collaboration with the cardiologist, as well as with other health providers that treat an AIT candidate, is recommended. Antihypertensive therapy should preferably be adapted to the relative guidelines [5,7,27]. However, it is not necessary to refer patients to their cardiologists before initiating AIT; it is only highly suggested for patients that have experienced sting-induced anaphylactic shock. In VIT candidates with a history of a severe reaction involving the cardiovascular system, a cardiological exam may detect a subclinical artery disease that can be deteriorated after VIT-induced anaphylaxis.

## 4. Neoplasias

According to the latest guidelines of the EAACI on AIT, malignant neoplasias are considered an absolute contraindication, but VIT is a highly advised option in venom-allergic patients with a life-threatening history [5,7,40]. The concern of allergologists is that AIT might stimulate tumor growth, even though the pathogenic impact of AIT in cancer is not well understood [40]. Furthermore, no controlled studies on the effectiveness or risks of AIT in these patients are available [40].

Cancer is a common disease that seriously threatens human health and is a major public-health problem worldwide [41]. Due to the characteristics of malignant tumors, such as limitless replicating potential, metastasizing capacity, immune-escaping ability and heterogeneity, the conventional diagnostic and therapeutic strategies are seriously challenged [41]. Moreover, due to the lack of sensitivity and specificity, most of the traditional diagnostic methods have limitations in the clinical application of an early cancer diagnosis. This is the reason why a detailed clinical examination of all systems and a thorough individual and family health history are crucial in assessing the patient for a possible malignancy.

Known factors that can influence the expression of cancer and require vigilance by the specialist who tries to detect high risk patients are race, sex, age and many environmental factors such as smoking, air pollution, drugs, diet and several pathogens. Certain genetic conditions and immune-deficiency syndromes are associated with an increased risk of developing cancer [42]. The family history (including cause of death) should include information about parents, siblings, and first cousins [43,44]. Age, serious illnesses, and congenital anomalies should also be elicited. Any of the above factors can point to a specific screening test for the early detection of cancer and/or successful exclusion from AIT.

Cancer is often difficult to detect in its early stages since the relevant signs and symptoms are nonspecific, with insidious onset, and can mimic other common disorders. Common signs and symptoms that may lead to the clinical suspicion of cancer are [45,46]:Unexplained paleness and loss of energyUnusual lump, mass, or swellingSudden unexplained weight lossUnexplained persisting fever or illnessEasy bruising or bleedingProlonged or ongoing pain in one or more areas of the bodyLimpingFrequent headaches, particularly in the morning and associated with vomitingSudden eye or vision changesUnexplained changes in bowel and urination habitsObvious changes in existing skin lesionsPersistent cough or hoarseness

A *thorough physical examination* includes several systems: Observation for general health appearance, central and peripheral skin color, nutritional status, respiratory rate and effort, sweating, venous distention, and edemaExamination of the chest wallExamination of the breasts, including assessment of pubertal stage in females and assessment for gynecomastia in males.Examination of the lungsExamination of the heart including palpation and auscultationExamination of the abdomenExamination of nose, mouth and neck

The workup of a patient with suspected cancer should be individualized based on the findings from the patient’s history and physical examination [45,46].

When the history and physical examination do not indicate a likely diagnosis, a basic diagnostic evaluation should include a CBC with a differential. A chest radiograph is highly advised, while it is optional to check electrolytes, glucose, calcium, renal and hepatic function, thyroid-stimulating hormone, erythrocyte-sedimentation rate (ESR) or C-reactive protein (CRP), urinalysis, stool for occult blood, and perform age-appropriate cancer screening (Table 1). Further evaluation should be based only on the results of these initial tests.

Table 1 addresses cancer screening for ages younger than 65 years old, which constitute the vast majority of AIT candidates. For elder candidates, their individual preferences, life expectancy and potential procedural complications of cancer-screening tests should be taken under consideration. Furthermore, some screening tests are suggested to be performed more frequently in elder persons; annually for the fecal occult blood test and CT scan for lung cancer, biennially for mammography, and pap smears may be safely discontinued after three consecutive normal test results in a ten-year period [46].

## 5. Autoimmune Rheumatic Diseases

There are scarce data to guide allergologists regarding the evaluation of immunotherapy candidates for possible autoimmune rheumatic disease (ARD). A retrospective cohort study showed that patients with allergic diseases appear to have a greater risk of developing ARDs, particularly systemic lupus erythematosus and Sjogren’s syndrome [47]. In addition, a recent meta-analysis examined the risk of the development of autoimmune diseases, including ARDs, in patients with atopic dermatitis and found that these patients have a greater risk of developing rheumatoid arthritis and systemic lupus erythematosus [48]. On the other hand, a previous study reported that patients with rheumatoid arthritis, the most common ARD, have a decreased prevalence of allergic diseases [49]. Furthermore, a case-control study showed that patients with rheumatoid arthritis and ankylosing spondylitis have a similar risk of developing allergic diseases compared to healthy controls [50].

A thorough history is crucial in assessing patients for possible ARD, with particular emphasis on obtaining a detailed review of systems. Even though the information obtained in the history can rarely point to a specific diagnosis, it allows the clinician to narrow the differential diagnosis.

The assessment for arthritic symptoms is of paramount importance given that most rheumatic-disease patients exhibit musculoskeletal manifestations such as arthralgia or arthritis, which is characterized by joint pain, swelling, and less often with the erythema. Features of the inflammatory pain include prolonged morning stiffness that usually lasts for more than 30 min, and the alleviation of symptoms with activity and exacerbation with immobility.

Extra-articular symptoms such as fatigue, alopecia, rash, and adenopathy are common in a wide variety of ARDs. Further, proximal muscle weakness, a common complaint in patients with idiopathic inflammatory myositis, can be assessed by asking if the patient experienced difficulty brushing his/her hair, rising from a sitting position or climbing stairs.

Patients should be queried about the presence of sicca symptoms, including dryness of the eyes and mouth and a foreign-body sensation in the eyes, which are features suggestive of Sjogren’s syndrome. Further, the presence of recurrent oral ulcers, often painless, particularly in the soft or hard palate, might be consistent with systemic lupus erythematosus. Painful oral or tongue ulcers that occur more than 3 times a year and might coexist with genital ulcers are characteristics of Behcet’s disease. Raynaud’s phenomenon is a relatively common symptom in ARDs, particularly in systemic sclerosis, which is characterized by a tricolor sequence of color changes (white to blue to red); however, not every patient experiences the classic full sequence.

On physical examination, special attention should be paid to the presence of synovitis, which is characterized by bogginess on the joint palpation, tenderness, limited joint motion, and increased warmth. The presence of synovitis, skin thickness and tightness of the joint palpation raises the possibility of an ARD. Muscle strength should be assessed using the muscle strength of the distal and proximal muscles. A comprehensive examination is essential and the identification of signs such as parotid enlargement, malar rash, skin lesions, lymphadenopathy, mucosal ulcerations, or bibasilar crackles may suggest an ARD.

After a physical examination that revealed characteristic signs and symptoms of ARD, serologic testing may be indicated [51]. For example, given the high prevalence of low titers of positive antinuclear antibody (ANA) in the healthy general population, testing every patient might lead to false-positive test results, further unnecessary workups and incorrect diagnoses [52]. The initial diagnostic workup should include complete blood count, renal- and liver-function tests, urinalysis. The sedimentation rate and C-reactive protein are nonspecific markers of inflammation but may be useful in differentiating between ARD and other non-inflammatory conditions. The ANA test is a sensitive test that detects ARD but has low specificity; therefore, a positive ANA (titer > 1:80) should be followed by additional serologic tests, such as anti-Ro (SSA), anti-La (SSB), RNP, ds-DNA, Smith, scleroderma-70 and centromere antibodies. In patients with features of inflammatory arthritis, the rheumatoid factor and the cyclic citrullinated peptide antibodies can be ordered to evaluate for possible rheumatoid arthritis.

## 6. HIV Infection

Most international guidelines and position statements consider HIV infection to be a relative contra-indication for allergen immunotherapy [10], whereas some others list acquired immunodeficiency as an absolute contraindication [53]. In the past, AIT was avoided in HIV-positive patients because of potential effects on the activation of infected CD4+ cells, resulting in viral proliferation and disease progression. This putative negative impact of AIT on HIV infection is purely theoretical. Since its introduction for the treatment of HIV, highly active antiretroviral therapy (HAART) has improved the immune function and life expectancy. It appears that HIV^+^ patients under HAART can be safely treated with all types of AIT [27].

On the contrary, AIT is contraindicated to patients with Acquired Immunodeficiency Syndrome (AIDS) stages of the infection [27]. In a web-based survey among members of the American Academy of Allergy and Clinical Immunology (AAACI) about their experience with SCIT in patients with certain medical conditions, AIDS was one of the three diseases where SCIT resulted in major problems, such as the activation of an underlying disease or systemic reactions [54]. A total of 25% and 10% of the participants’ allergists performed SCIT in 420 HIV-positive allergic individuals and 179 patients with AIDS, respectively. Major problems resulting in the discontinuation of SCIT appeared in 4.2% of patients with AIDS and 0.9% of HIV-positive individuals, but the survey did not explore whether the problems were due to activation of the underlying disorder or from intolerance to SCIT. A similar study with VIT showed higher rates of major problems related to the safety of VIT in its use in patients with AIDS but not in patients in the initial stages of HIV infection [55]

Recent guidelines on AIT’s contraindications state that AIT can be performed on an individual basis in HIV-positive patients under HAART who have no severe symptoms (Categories A and B/CDC 1993 classification) [56] and with current CD4^+^ > 200 cells/μL [27].

As antiretroviral therapy is recommended to start as soon as possible for all individuals with HIV and detectable viremia, in HIV positive patients starting AIT, the viral load and CD4^+^ cell count should be monitored at the time of entry into AIT, monthly for the first three months and then every three months if the viral load remains consistently suppressed. More-frequent monitoring should be conducted in the case of clinical symptoms connected to the infection [57].

Since HIV detection can offer the chance to promptly start HAART and avoid a disease progression, it is highly suggested to patients starting AIT. The connection of allergen immunotherapy with immunomodulation mechanisms in HIV disease remain to be clarified. It is not clear whether the activation of infected CD4^+^ cells in patients under sublingual immunotherapy is similar to the effect of subcutaneous AIT.

## 7. Pregnancy

Although it is recommended to cautiously continue a well-tolerated AIT during pregnancy, its initiation is contraindicated in order to avoid the consequences of an anaphylactic episode [27,58]. Therefore, a possible pregnancy should be excluded through detailed, physical examination and, if needed, laboratory tests before starting AIT. The woman should describe her usual menstrual pattern, including the date of onset of the last menses and their frequency. The classic presentation of pregnancy is a woman with menses of regular frequency who presents with amenorrhea, nausea, vomiting, generalized malaise and breast tenderness. Information that may suggest early signs of pregnancy are an atypical last menstrual period and/or irregular menses and the fact that about 25% of women bleed during their first trimester. To confirm or exclude the diagnosis, the beta-subunit of human chorionic gonadotropin (β-hCG) can be measured in maternal serum and urine.

## 8. Chronic Diseases

There is a large heterogeneity among guidelines regarding the inclusion of immunodeficiencies and chronic infectious diseases as AIT contraindications [6,10]. There is no solid evidence in the literature that chronic infections such as hepatitis B or C might be affected by AIT, so they are considered as relative contraindications for all routes and types of AIT [27]. The doubtful effect of AIT on the health of a subject with an undiagnosed chronic disease suggests that hepatitis serology markers are optional. Their monitoring as precautionary measures protecting AIT-performing health-providers is not justified, since needles should always be handled with caution, whether managing contagious patients or not.

In patients with sarcoidosis, there is the possibility that the inoculation of injected antigens during AIT induces granulomatous lesions at the vaccination site [59]. However, the possibility of a sarcoidosis relapse should only be considered if sarcoid granulomas appear at the inoculation site of the injections, making re-evaluation necessary.

Primary immunodeficiencies have been proposed as contraindications for AIT, mainly due to the concern of the limited efficacy of the treatment [27,60]. No evidence of any harmful effect of AIT exists in the case of immunodeficiencies or in patients under immunosuppressive drugs. If any chronic disease is reported, close monitoring should be continued during AIT and the efficacy and safety of the procedure should be regularly evaluated.

## 9. Asthma

Allergologists are familiar with the diagnosis and treatment of asthma and it is self-evident that spirometry is regularly performed in patients with asthma, or with symptoms suggesting it. Starting any kind of AIT in patients with uncontrolled asthma is absolutely contraindicated, while patients with partially controlled asthma may be treated with AIT [6,27]. Allergoids and SLIT, which have a safer profile than natural depot extracts, have been proposed for the treatment of severe allergic asthma (although SLIT’s efficacy is questionable in severe asthma) [8].

A modest increase in the risk of adverse events is associated with SCIT and SLIT administered for the treatment of allergic asthma [1]. In order to minimize adverse events, as clearly suggested by the EAACI guidelines, “the level of ‘current clinical control’ for each patient should be properly assessed, measuring peak flow before each injection and postponing the injection if lung function has decreased > 20% of the personal best value” [27].

## 10. Conclusions and Unmet Needs

According to the Hippocratic precept, “in illnesses one should keep two things in mind; to do good or to do no harm”. The intention of the physician that prescribes a medicine is to cure, taking care to minimize the manifestation of unwanted adverse reactions. The same intention applies to the administration of AIT that aims to “correct” the way that the human immune system reacts to innocuous antigens, but is avoided if a concomitant disease or health condition may be deteriorated.

The use of AIT is a precision-medicine approach for the treatment of respiratory and venom allergy in adults and children. The patient’s thorough anamnesis should be taken before the start of AIT, since the description of clinical symptoms and signs may offer clues suggestive of a concomitant disease, which could affect treatment decisions. For example, the mention of symptoms suggestive of eosinophilic esophagitis discourages the use of SLIT to airborne allergens, while SCIT is considered safer [61].

Allergologists should always be alert to any change in patients’ general health condition throughout the AIT duration. The periodic performance of peak-flow measurement and spirometry is a familiar example of periodical follow-up practice for AIT-treated patients with asthma. Therefore, during AIT visits, besides asking about their allergy symptoms, it is also suggested to ask patients about the general condition of their health.

A laboratory workup or a reference to an expert can help to define the diagnosis of suspected diseases. On the other hand, ordering an extended workup without proper justification can be considered as thriftlessness. The cost of an extensive evaluation that is not always approved and covered by public or private insurance providers is a parameter that should be taken under consideration. Proper decision making and considering the impact of each laboratory test sets the basis of cost-effectiveness. However, not all benefits and costs (transportation, out-of-pocket expenses and productivity losses) are health related, so it would be wise to approach patients from a societal perspective as well [62].

Although as physicians we are not always well trained to make cost-effectiveness decisions, we are at least trained to the individual assessment of the risk-benefit ratio; we can decide on the risk vs. benefit when ordering a chest X-ray or further cancer-imaging tests, which expose our patient to radiation [63].

In the present paper, it was attempted to approach the dilemma of what makes a test or examination helpful before the start of AIT, in the hope of triggering subsequent studies. In Figure 1, a “decision tree” is proposed. It appears that when the history and physical examination are not conclusive of a safe diagnosis, further investigation is required. In Table 2, an approach on how to proceed with the preliminary workup for AIT candidates is suggested mainly based on the guidelines on contraindications that are currently in effect [5,6,7,27].

Since there is no gold standard for screening tests, physicians should (or should not) order them based on a tailor-made approach to each AIT candidate, taking under consideration contraindications and the predictive value of each test. Undoubtedly, more research is required in order to establish a universal approach.

## Figures and Tables

**Figure 1 cells-11-00653-f001:**
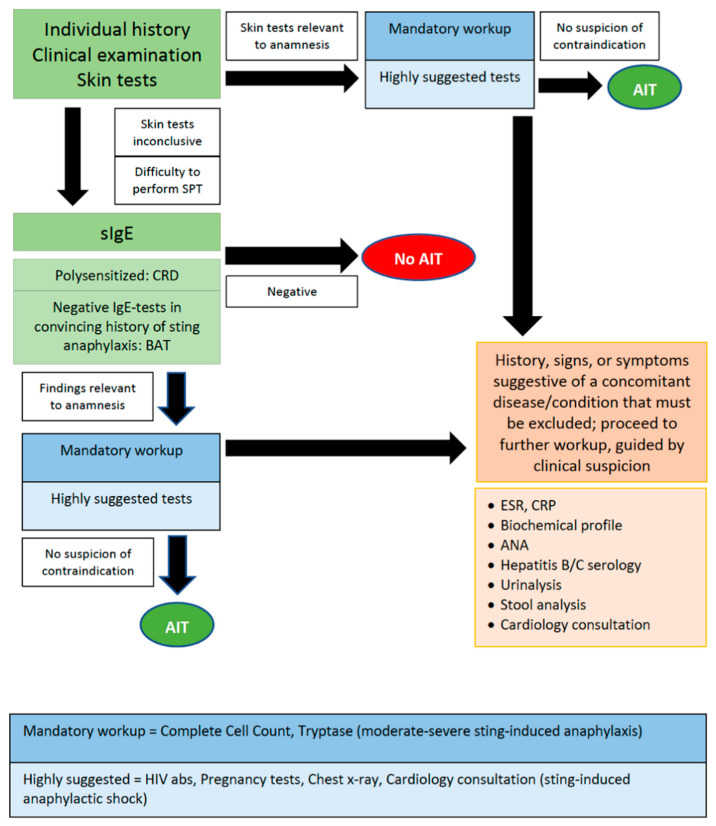
Suggested diagnostic procedure before the start of AIT.

**Table 1 cells-11-00653-t001:** Screening and prevention for adults younger than 65 years old.

Cancer		Action
Breast cancer	Concerning family history	Refer for genetic counseling/testing
	Hereditary breast and ovarian syndrome	Screen per recommendations
	Women > 40	Individual decision; if screening desired, screen with mammography every two years
Cervical cancer	Women 21 to 29 years	Pap smear every three years
	Women ≥ 30 years	Pap smear every three years, or Pap smear + HPV testing every five years
Colorectal cancer	Patients with risk factors	Screen per recommendations
	Patients ≥ 50 years without risk factors	Screening (decide among colonoscopy, flexible sigmoidoscopy, fecal occult blood test)
Lung cancer	Patients 55 to 74 years, ≥30 pack-year smoking history and either currently smoking or quit in the past 15 years	Consider screening with low-dose helical CT scan
Prostate cancer	High-risk men 40 to 45 years	Discuss screening, individual decision
	Men ≥ 50 years without risk factors	Discuss screening, individual decision
Melanoma	High-risk patients	Periodic skin exam
	Average-risk patients	Remain vigilant for suspicious lesions

**Table 2 cells-11-00653-t002:** Suggestions on how to proceed with workup in candidates for AIT, after having detected the relevant symptom-developing allergen.

Workup	Suggestions
*sIgE and total IgE*	Optional
*Molecular sIgE (CRD)*	Optional (polysensitized patients)
*Tryptase*	VIT; mandatory in moderate-severe sting-induced anaphylaxis
*BAT*	VIT; optional (exceptional cases of negative IgE-tests)
*Complete cell count*	Mandatory
*Glucose, BUN, creatinine, AST, ALT, albumin, electrolytes*	Optional
*ESR, CRP*	Optional
*HIV detection*	Highly suggested
*Hepatitis B/C serology*	Optional
*ANA*	Highly suggested, only when physical examination reveals characteristic signs and symptoms posing probability of ARD
*Pregnancy test*	Highly suggested in atypical last menstrual period and/or irregular menses
*Urinalysis*	Optional
*Stool analysis*	Optional
*Chest X-ray*	Highly suggested
*Cardiology consultation*	AIT; optional, when cardiologic problems preexist. VIT; highly suggested in history of sting-induced anaphylactic shock.

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
