# Peer review of "Workup and Clinical Assessment for Allergen Immunotherapy Candidates"

_cells, 2022, doi:10.3390/cells11040653_

Round 1

Reviewer 1 Report

General comments

Authors review the use of allergen immunotherapy (AIT) in allergic patients suffering from concomitant diseases such as cancer, HIV, heart conditions and immunodeficiencies among others. They claim that there is a gap of knowledge and a lack of published studies examining the appropriate evaluation for AIT patients, with no globally accepted guidelines. While they are addressing important issues there are some concerns that need to be addressed. This review is based on a report that the authors have previously published (Clin Transl Allergy 2019;9:45)

Specific comments 

Tests confirming IgE-mediated allergy and…A complete list of allergy tests is given. However, there is no description of the molecular mechanisms by which AIT may cause side effects. The manuscript will improve by adding a section explaining inflammatory mechanisms induced by AIT administration (a figure could be drawn about this subject).

Cardiovascular checkup. authors should discuss clinical reports and provide references of cardiovascular alterations associated with AIT. what is the incidence of cardiovascular alterations induced by AIT?  Similarly, what is the incidence of AIT side effects on patients taking b-blockers, ACE inhibitors and other antihypertensive drugs.
Reference 23 is not related to cardiovascular manifestations of anaphylaxis.

Neoplasias. Like in the cardiovascular disease, there is not description of the incidence of allergic disease in cancer patients. Similarly, autoimmune rheumatic diseases (ARD) are described. However, there is not any description of the allergic diseases previously associated to ARD. In contrast, good information about allergic disease related to HIV was provided.

Autoimmune rheumatic diseases (ARD). There is almost nothing mentioned about allergic disease in ARD. Only a short statement was made “There are scarce data to guide allergologists regarding ARD” The remaining text is focused on describing symptoms as well as diagnostic approaches for ARD. Are there any reports ARD in allergic patients?

Asthma symptoms are some of the most frequent side effects of AIT, particularly in severe allergic asthma. However, this disease is not discussed in this manuscript. A new subtitle “lung disease” could be added and include asthma.

The title of the manuscript could be more specific: For example, “AIT in allergic patients suffering from concomitant diseases”

Author Response

Following your advice we changed the title of the manuscript. However, the aim of this article was to suggest a preliminary workup to patients that have already been diagnosed with allergy and are candidates for AIT. A preliminary workup may exclude "contraindicated" existing concomitant diseases/conditions in apparently healthy subjects. We have made relevant changes in the abstract too, since the purpose of our paper might not have been very obvious.

Following your suggestions, we have made changes regarding the paragraph on cardiovascular checkup, to better clarify the relevance of beta-blockers and ACE-inhibitors with AIT, we have described data on incidence of allergic diseases in patients with rheumatic diseases and we have added a paragraph on asthma. However, we are also afraid that this is not a general review on AIT, including the underlying mechanisms, so we considered that the description of AIT mechanisms would be a pleonasm, beyond the practical clinical issue we are addressing. Allergy and cardiovascular diseases or cancer are not inter-connected in pathophysiology and a discussion on their common incidence would make our text prone to unwanted debates.

Thank you for underlying the mistake of Reference 23. Apparently, there was a mistake in all references of this paragraph and we corrected it.

Reviewer 2 Report

I suggest reducing the extent of chapter 4 (Neoplasms). On the other hand, Table 1 could summarize many concepts already expressed.

Author Response

We have reduced the paragraph on neoplasias

Reviewer 3 Report

Please read the attached file: To Authors.docx

Author Response

Thank you for your suggestion. It is apparently interesting to address the question you are posing with a new article entitled “Is allergen immunotherapy still relevant today?”. We are sorry that you have misgivings about recommending AIT for seasonal allergic rhinitis.

You are asking on our beliefs on whether a patient should choose allergen immunotherapy and you are supporting that pharmacotherapy is cheaper and more effective, but studies of cost-effectiveness do not sustain it. But most importantly, this issue is completely irrelevant with our article.

AIT is considered an evidence-based therapy for allergic rhinitis (including seasonal) and we tried to address the need of a preliminary workup in order to avoid the treatment of patients that present undiagnosed contraindicated conditions, potentially affected by AIT. Apparently, your criticism refers to all articles on allergen immunotherapy provided for seasonal allergic rhinitis and/or asthma.

We would appreciate your independent look on the issue we are addressing.

Reviewer 4 Report

The proposed manuscript suggests an approach on the workup and the assessment for the presence of the underlying diseases/conditions, before starting AIT. The manuscript is well written and easy to follow, summarizing the major diseases/conditions that must be considered before suggesting the administration of an AIT to specific patients. Additionally, it seems well structured, presenting most relevant and updated literature. The manuscript could be of potential publication in Cells journal, after addressing the following comments.

Comments:

Introduction - should also include information from EAACI Allergen Immunotherapy User's Guide, recently published. Please consult https://doi.org/10.1111/pai.13189.

Line 45 - please discriminate the target AIT, respiratory? Venom?

Lines 84-91 - The text should be under the section of SPT.

Section CDR - please provide clear definitions for cross-reactivity, cross-sensitivity, polysensitization, co-sensitization, since some of these terms are used with the same synonym, which can cause some confusion. Define the criterion for their use in the manuscript.

Line 108 - IgE is by definition an antibody, therefore IgE antibody is redundant. Delete "antibody".

Line 201 - correct to "…even if a history…"

Line 210 - provide acronym for ACE.

The manuscript would also benefit (although not mandatory, but highly recommended) from a figure with a tentative “decision tree” as the workup summarizing the information gather in the different sections of the diseases/conditions and whether a patient should follow to AIT or not. Please consider to add the requested fig.

Author Response

Thank you very much for your kind remarks. Although we are aware of the EAACI AIT user’s guide we haven’t included it since most points described in it and relevant to our paper were included in other EAACI position papers. Nevertheless, we have included it now in several points of the text and of course we used it as a reference enforcing the fact that our text is also addressing paediatric AIT. We also incorporated your other suggestions and made the appropriate corrections. In detail:

-You were absolutely right to ask us to discriminate the target of adverse events (Line 45). Instead of writing subcutaneous AIT for VIT and SCIT, we preferred to be more descriptive on the incidence of adverse reactions.

- Line 84-91. Added a SPT section

-CDR section. Thank you for your excellent notice. We have deleted the term cross-sensitivity that was confusing and inappropriate. In order to explain the meaning of clinically relevant cross-reactivity we have tried to explain in one sentence introducing the term polyallergic and differentiate it to polysensitive patient.

-Line 108, 201; corrected.

-Line 210: provided acronym

Added a figure regarding a “decision tree”

Round 2

Reviewer 1 Report

The manuscript has improved significantly. It can be accepted for publication now.

Reviewer 3 Report

None